# Interleukin-26 Has Synergistic Catabolic Effects with Palmitate in Human Articular Chondrocytes via the TLR4-ERK1/2-c-Jun Signaling Pathway

**DOI:** 10.3390/cells10092500

**Published:** 2021-09-21

**Authors:** Yi-Ting Chen, Chih-Chien Wang, Chia-Pi Cheng, Feng-Cheng Liu, Chian-Her Lee, Herng-Sheng Lee, Yi-Jen Peng

**Affiliations:** 1Graduate Institute of Medical Sciences, National Defense Medical Center, Taipei 114, Taiwan; z987122@gmail.com; 2Department of Orthopedics, Tri-Service General Hospital, National Defense Medical Center, Taipei 114, Taiwan; tsghcc@yahoo.com.tw; 3Department and Graduate Institute of Biology and Anatomy, National Defense Medical Center, Taipei 114, Taiwan; ph870317@ndmctsgh.edu.tw; 4Rheumatology/Immunology and Allergy, Department of Medicine, Tri-Service General Hospital, National Defense Medical Center, Taipei 114, Taiwan; lfc10399@yahoo.com.tw; 5Department of Orthopedics, School of Medicine, College of Medicine, Taipei Medical University and Hospital, Taipei 114, Taiwan; chianherlee@yahoo.com.tw; 6Department of Pathology and Laboratory Medicine, Kaohsiung Veterans General Hospital, Kaohsiung 813414, Taiwan; hlee@vghks.gov.tw; 7Department of Pathology, Tri-Service General Hospital, National Defense Medical Center, Taipei 114, Taiwan

**Keywords:** IL-26, palmitate, human articular chondrocytes, inflammatory arthritis

## Abstract

The inflammatory cytokine interleukin-26 (IL-26) is highly expressed in the serum and synovial fluid of patients with inflammatory arthritis. The effect of IL-26 on human articular chondrocytes (HACs) remains unclear. Obesity is associated with disability of patients with rheumatoid arthritis and disease activity in those with ankylosing spondylitis. The saturated free fatty acid palmitate with IL-1β can synergistically induce catabolic effects in HACs. The aim of this study was to evaluate the effects of IL-26 and palmitate in HACs. In this study, palmitate markedly synergizes the IL-26-induced proinflammatory effects and matrix protease, including COX-2, IL-6, and MMP-1, in HACs via the toll-like receptor 4 (TLR4)-ERK1/2-c-Jun signal transduction pathway. The synergistic catabolic effects of palmitate and IL-26 were attenuated by inhibitors of TLR4 (TAK242), ERK1/2 (U0126), or c-Jun (SP600125) in HACs and cartilage matrix. In addition, metformin, a potential inhibitor of TLR4, also decreased expression of COX-2 and IL-6 induced by co-incubation with IL-26 and palmitate. IL-26 and palmitate synergistically induced expression of inflammatory and catabolic mediators, resulting in articular cartilage matrix breakdown. The present study also revealed a possible mechanism and therapeutic targets against articular cartilage degradation by increased saturated fatty acids in patients with inflammatory arthritis.

## 1. Introduction

Interleukin-26 (IL-26) is a proinflammatory cytokine, initially identified in herpesvirus-transformed T cells in 2000 [1], that can induce the production of other proinflammatory cytokines via activation of its heterodimer receptor, a complex of IL-10 receptor 2 and IL-20 receptor 1, through phosphorylation of signal transducers and activators of transcription (STAT)-1/3 and extracellular signal-regulated protein kinases (ERK)-1/2 [2]. The induction of IL-26 has been reported in many inflammatory diseases, including Crohn’s disease [3], chronic obstructive pulmonary disease [4], and sepsis [5]. IL-26 expression is increased in the serum and synovial fluid of patients with inflammatory arthritis, rheumatoid arthritis, spondylarthritis, and psoriatic arthritis [6]. However, the direct effects of IL-26 on human articular chondrocytes (HACs) remain unclear.

Obesity is a global public health concern resulting in comorbidities of coronary artery disease, hypertension, type 2 diabetes, and osteoarthritis [7]. Obese individuals have higher circulating levels of free fatty acids (FFAs) produced by adipocytes [8]. Saturated FFAs can induce cytokine production via activation of toll-like receptor (TLR) 2/4 through mitogen-activated protein kinase (MAPK) and the nuclear factor (NF)-κB or AP-1 signaling pathway [9,10,11]. A previous study reported that the saturated FFA palmitate (C16:0), but not monounsaturated oleate (C18:1), has synergistic catabolic effects with IL-1β in HACs [12]. High rates of obesity are also associated with disability of patients with rheumatoid arthritis and disease activity of ankylosing spondylitis [13,14]. However, the association of IL-26 and saturated FFAs in HACs has not been investigated. Therefore, the aim of the present study was to evaluate the synergistic effects of palmitate and IL-26 in HACs.

## 2. Materials and Methods

### 2.1. Reagents

Recombinant human IL-26 protein (MBS1362134) was purchased from MyBiosource (San Diego, CA, USA). Palmitate (P0500), metformin (D150959), methylthiazolyldiphenyl-tetrazolium bromide (MTT) (M5655), and bovine serum albumin (BSA) (A7030) were purchased from Sigma-Aldrich Corporation (St. Louis, MO, USA). Antibodies against cyclooxygenase-2 (COX-2; 1:1000; 35-8200) were obtained from Thermo Fisher Scientific (Waltham, MA, USA), while those against collagen type II (COL-II; 1:1000; ab34712) and matrix metalloproteinase-1 (MMP-1) (1:1000; ab134184) were purchased from Abcam (Cambridge, MA, USA), and those against IL-6 (1:1000; #12153), Erk1/2 (1:1000; #4695), phospho-Erk1/2 (1:1000; #9101), phospho-c-Jun (1:1000, #3270), phospho-p38 (1:1000, #9211), phospho-JNK (1:1000, #9251), STAT1 (1:1000, #14994), phospho-STAT1 (1:1000, #9131), STAT3 (1:1000, #9139), phospho-STAT3 (1:1000, #9131), histone H3 (1:2000, #9715), and glyceraldehyde 3-phosphate dehydrogenase (GAPDH; 1:2000; #5174) were purchased from Cell Signaling Technology, Inc. (Danvers, MA, USA).

### 2.2. Isolation and Culture of HACs

Human articular cartilage specimens were obtained from patients who underwent total knee replacement in Tri-Service General Hospital (Taipei, Taiwan; TSGHIRB approval no. 1-102-05-091). The HACs were prepared as described in our previous report [15]. Briefly, the articular cartilage specimens were cut into small pieces that were immediately placed in antimicrobial solution containing 500 IU/mL of penicillin (Gibco, Carlsbad, CA, USA), 500 mg/mL of streptomycin (Gibco), and 2.5 mg/mL of Fungizone (Sigma-Aldrich Corporation) for 4 h, and then washed twice with sterile phosphate-buffered saline. HACs were extracted by sequential enzymatic digestion with 0.25% trypsin (Gibco) and collagenase type H:type II (1:1) (Sigma-Aldrich Corporation), and then incubated at 37 °C under a humidified atmosphere of 5% CO_2_/95% air. The experimental primary culture cells were passaged 3 to 5 times.

### 2.3. Palmitate/BSA Complex Solution

Palmitate (C16:0) was dissolved in 0.6% alcohol and 0.01 M sodium hydroxide by vortexing, heated to 75 °C for 10 min, and then conjugated with 1% non-esterified fatty acid (NEFA)-free BSA in Dulbecco’s modified Eagle’s medium–nutrient mixture F-12 (DMEM/F-12) to a final concentration of 1 M in palmitate. The palmitate/BSA complex solution was sterilized using membrane filters with a pore size of 0.45 μm. Serum-free medium containing 1% NEFA-free BSA and 0.6% alcohol was used as a vehicle control. Each palmitate stock solution was prepared as previously described [16].

### 2.4. Cell Viability

HACs were seeded at 1 × 10^4^ cells in the wells of 96-well plates and treated with palmitate (0.25–1.0 mM) alone or in combination with IL-26 (100 ng/mL) for 24 h in an incubator under an atmosphere of 5% CO_2_/95% air at 37 °C. MTT solution (2 mg/mL) was added to each well for 3 h followed by dimethyl sulfoxide for 10 min. The optical density at a wavelength of 570 nm of each well was determined with a microplate reader.

### 2.5. Cell Stimulation and Immunoblotting

HACs were seeded at 5 × 10^5^ cells in 60 mm cell culture Petri dishes. The confluent monolayer cells were incubated in serum-free medium for 2 h before treatment with 0.25 mM BSA with conjugated palmitate and IL-26 (100 ng/mL). Before treatment with palmitate and IL-26, the cells were pretreated with inhibitors for 1 h. After incubation, the cells were washed with sterilized ice-cold phosphate-buffered saline and lysed with radioimmunoprecipitation assay buffer containing a protease inhibitor cocktail (Roche Diagnostics, Mannheim, Germany). Whole cell lysates were centrifuged at 14,000 rpm for 15 min to remove all insoluble materials. The protein concentrations were detected using the bicinchoninic acid assay (Thermo Fisher Scientific). Equal amounts of proteins were separated by 10% sodium dodecyl sulfate–polyacrylamide gel electrophoresis and then transferred to polyvinylidene fluoride membranes (EMD Millipore Corporation, Billerica, MA, USA), which were blocked with 1% bovine serum albumin–Tris-buffered saline with Tween 20 (TBST) for 1 h at room temperature. After washing with TBST, the membranes were incubated overnight with primary antibodies at 4 °C and then washed six times before incubation with horseradish peroxidase-labeled secondary antibodies (1:10,000; Jackson ImmunoResearch Laboratories, West Grove, PA, USA) for 1 h. After probing for signaling proteins, we visualized bands on the membranes using an enhanced chemiluminescence reagent. All immunoblotting experiments were repeated at least three times.

### 2.6. Extraction of Cytoplasmic and Nuclear Protein

Cytoplasmic and nuclear extracts were prepared using the Extract-EZ G, Nuclear and Cytoplasmic Protein Extraction Kit (Bio Basic Inc., Markham, ON, Canada) in accordance with the manufacturer’s protocol. In brief, the cell pellet was resuspended in ice-cold reagent A and vortexed for 10 s per 20 min of incubation on ice. After incubation, the contents of the tube were mixed and centrifuged at 12,000× *g* for 10 min. Then, the supernatant (cytoplasmic fraction) was collected. The cell pellet, which contained the nuclei, was resuspended in reagent B, vortexed for 10 s per 20 min of incubation on ice, and centrifuged at 12,000× *g* for 10 min. Afterward, the supernatant (nuclear fraction) was collected. The purity of the cytoplasmic and nuclear fractions was then assessed by immunoblotting for GAPDH and histone H3, respectively.

### 2.7. Preparation of Cartilage Explants

Selected articular cartilage specimens with intact surfaces (grade 0 to 1 of the Collins/McElligott system [17]) were cut into 6-mm^2^ pieces, weighed, and transferred to the wells of a 24-well plate and cultured for 24 h in DMEM/F12 containing antibiotics and 10% fetal bovine serum. After resting for 72 h in serum-free DMEM/F12, the cartilage explants were used for further study.

### 2.8. Staining with Safranin O and 1,9-Dimethylmethylene Blue for Analysis of Cartilage Degradation

The extent of degradation of the cartilage explants was determined by staining with Safranin O and fast green and measuring the amount of sulfated glycosaminoglycan (GAG) released from the culture medium. A complex of 1,9-dimethylmethylene blue (DMMB) solution (Sigma-Aldrich Corporation) and GAG was used to determine the amount of proteoglycan released in the medium, as an indirect indicator of cartilage degradation [18]. The amount of the GAG–DMMB complex formed was determined with a microplate reader at a wavelength of 595 nm. Finally, the total amount of GAG released per milligram of cartilage was calculated to determine the loss of cartilage in each group (*n* = 6). The cartilage explants were also fixed in formalin, embedded in paraffin, and cut into sections, which were stained with Safranin O/fast green and counter stained with Weigert’s iron hematoxylin to evaluate changes in proteoglycan content.

### 2.9. Statistical Analysis

Quantitative data are presented as the mean ± standard deviation (SD). Differences between groups were assessed with Student’s *t*-test. A probability (*p*) value of <0.05 was considered statistically significant.

## 3. Results

### 3.1. Cytotoxicity of IL-26 and Palmitate in HACs

The MTT assay was used to assess the cytotoxicity of palmitate with/without IL-26 on HACs by testing various concentrations of palmitate (0.25, 0.5, and 1.0 mM) for 24 h. IL-26 at a concentration of 100 ng/mL had no effect on the proliferation of HACs within 24 h, consistent with our previous study with the use of macrophages [19]. Palmitate at a concentration of >0.5 mM and at 0.25 mM combined with 100 ng/mL of IL-26 had no significant cytotoxic effect on HACs (Figure 1). Thus, 0.25 mM palmitate was adopted for the following experiments.

### 3.2. Effects of IL-26 and Palmitate on HACs

The effects of IL-26 and palmitate on catabolic and anabolic factors in HACs were evaluated. The expression levels of COX-2, MMP-1, IL-6, and COL-II in HACs treated with palmitate with/without IL26 was assessed by Western blot analysis. IL-26 induced the overexpression of COX-2 by 3.84 ± 1.043-fold as compared with control cells (*p* = 0.035). Palmitate also increased COX-2 expression by 2.053 ± 0.574-fold, but this increase was not statistically significant as compared with the control cells (*p* = 0.116). However, the combination of IL26 and palmitate significantly increased COX-2 expression by 10.59 ± 2.089-fold as compared with the control cells (*p* = 0.004), by 3.84 ± 1.043-fold as compared with IL-26 only (*p* = 0.028), and by 2.053 ± 0.574-fold as compared with palmitate only (*p* = 0.008). Palmitate and IL-26 individually significantly increased MMP-1 expression by 1.67 ± 0.19-fold (*p* = 0.017) and 6.37 ± 0.27-fold (*p* = 0.0002), respectively, as compared with the control cells. The combination of IL26 and palmitate significant increased MMP-1 expression by 15.77 ± 2.59-fold as compared with the control cells (*p* = 0.0013), and significantly increased expression by 2.14 ± 0.86-fold as compared with IL-26 only (*p* = 0.018) and 14.14 ± 2.6-fold as compared with palmitate only (*p* = 0.0016). Both palmitate and IL-26 individually significantly increased IL-6 expression by 1.55 ± 0.11-fold (*p* = 0.016) and 2.95 ± 0.6-fold (*p* = 0.017), respectively, as compared with the control cells. The combination of IL26 and palmitate significantly increased IL-6 expression by 12.95 ± 1.82-fold as compared with the control cells (*p* = 0.0004), while IL-26 and palmitate individually also significantly increased expression by 6.30 ± 1.92-fold (*p* = 0.0012) and 12.4 ± 1.82-fold (*p* = 0.0005), respectively. The combination of IL-26 and palmitate significantly decreased COL-II expression as compared with either palmitate or IL-26 alone (Figure 2). Collectively, the combination of IL-26 and palmitate might accelerate extracellular matrix destruction by markedly synergistically increasing the production of catabolic factors and decreasing of anabolic factors.

### 3.3. Effects of IL-26 and Palmitate on Signaling Pathways in HACs

Next, the effects of IL-26 and palmitate on signaling pathways in HACs were assessed. Previous studies have demonstrated that palmitate induced the production of inflammatory mediators via TLR4/MyD88-dependent mitogen-activated protein kinases (MAPKs) [20,21]. IL-26 is reported to induce the production of inflammatory mediators via the STAT1 and STAT3 signaling pathways [22]. The result showed IL-26 phosphorylated STAT1 and MAPKs, including ERK1/2, JNK, and p38, within 3 and 1 h in HACs, respectively, but not STAT3 (Appendix A). Palmitate only activated ERK1/2 signaling within 1 h, but not JNK, p38, STAT1, or STAT3 (Appendix A). Next, the effects of IL-26 or/and palmitate on ERK1/2, NF-κB, and c-Jun singling within 1 h were investigated. Co-treatment with IL-26 and palmitate significantly increased phosphorylation of ERK1/2 by 4.07 ± 0.27-fold as compared with the control group (*p* < 0.0001). Co-treatment with IL-26 and palmitate also significantly increased phosphorylation as compared to the addition of either IL-26 or palmitate to whole cell lysate (2.28 ± 0.38-fold (*p* = 0.001) for IL-26 vs. IL-26 and palmitate; 2.7 ± 0.29-fold (*p* < 0.0001) for palmitate vs. IL-26 and palmitate; Figure 3A). The combination of IL-26 and palmitate significantly increased phosphorylation of nuclear c-Jun by 6.21 ± 1.54-fold (*p* = 0.02) as compared with the control group, which was greater than either IL-26 or palmitate individually (4.35 ± 1.56-fold (*p* = 0.04) for IL-26 vs. IL-26 and palmitate; 4.42 ± 1.56-fold (*p* = 0.04) for palmitate vs. IL-26 and palmitate; Figure 3B). However, there was no significant change in phosphorylation of NF-κB (data not shown). Collectively, these results indicate that phosphorylation of ERK1/2 and c-Jun is greater with the combination of IL-26 and palmitate than either IL-26 or palmitate alone within 1 h. However, the relationship between these signaling pathways and catabolic mediators remains unclear.

### 3.4. Synergistic Catabolic Effects of IL-26 and Palmitate via the TLR4/ERK1/2-c-Jun Signaling Pathway

To determine whether the effects of palmitate and IL-26 involve the TLR4/ERK1/2/c-Jun signaling pathway, we pretreated HACs for 1 h with three inhibitors against TLR4 (TAK242), ERK1/2 (U0126), and c-Jun (SP600125). Co-treatment of HACs for 24 h with IL-26 and palmitate significantly increased the expression levels of COX-2 by 8.78 ± 1.46-fold, MMP-1 by 5.26 ± 0.74-fold, and IL-6 by 12.63 ± 2.27-fold. The results revealed that all of the tested inhibitors reduced the increased expression of COX-2, MMP-1, and IL-6 by co-treatment of IL-26 and palmitate in HACs (all, *p* < 0.05) (Figure 4).

Metformin is a first-line oral anti-blood glucose-lowering agent for type 2 diabetes [23]. A cohort study revealed that long-term administration of metformin might attenuate obesity in patients with arthritis symptoms [24]. Metformin can also potentially attenuate signaling of the TLR4 inflammatory pathway [25]. Therefore, the attenuating effect of metformin was investigated in HACs treated with the combination of IL-26 and palmitate. Briefly, HACs were pretreated with 5 mM metformin for 1 h and then co-treated with IL-26 and palmitate for 24 h. The results showed that metformin significantly attenuated expression of COX-2, MMP-1, and IL-6 via the TLR4-ERK1/2-c-Jun signaling pathway (Figure 4). Metformin also attenuated the catabolic effects, although the underlying mechanism remains to be determined.

### 3.5. Attenuation of Palmitate- and IL-26-Induced Matrix Degradation of Human Cartilage Explants by TAK242

To evaluate the catabolic effect of IL-26 and palmitate on articular cartilage, we employed Safranin O/fast green staining and the DMMB assay to assess the release of glycoproteins in articular cartilage. The results of the DMMB assay showed that IL-26 or palmitate alone did not increase the release of GAG from cartilage explants within 72 h. However, co-treatment with IL-26 and palmitate significantly increased the release of GAG by 1.75 ± 1.12-fold (*p* = 0.0056). However, the synergistic effect of IL-26 and palmitate was attenuated by pre-treatment with TAK242 for 1 h (Figure 5B). The results of the DMMB assay were consistent with Safranin O/fast green staining of the cartilage explants (Figure 5A). The ex vivo and in vitro results were also consistent.

## 4. Discussion

Inflammatory arthritis is associated with high serum and synovial fluid levels of IL-26, obesity, and disease activity [13,14]. This is the first report of the proinflammatory effects of IL-26 on articular cartilage and synergistic action with palmitate to promote the breakdown of the cartilage matrix.

In the present study, the effects of IL-26 and palmitate on HACs and cartilage explants were investigated by evaluation of inflammatory mediators and matrix proteinase. Either IL-26 or palmitate increased the expression levels of COX-2, IL-6, and MMP-1, but not COL-II. Moreover, as compared to palmitate, IL-26 significantly increased the expression levels of these proinflammatory mediators. Furthermore, co-treatment with IL-26 and palmitate synergistically increased expression of COX-2, IL-6, and MMP-1, and decrease expression of COL-II. Furthermore, IL-26 and palmitate degraded cartilage explants. Collectively, these results demonstrated the synergistic catabolic effects of IL-26 and palmitate in articular cartilage.

The pathophysiologic effects IL-26, a member of the IL-10 family, remain to be fully elucidated [26]. IL-26 is reported to activate host defenses against viruses and bacteria by priming of immune cells, including neutrophils, natural killer cells, and plasmacytoid dendritic cells, and by forming pores in the bacterial membrane [27,28]. In response to IL-26, monocytes and respiratory epithelial cells secrete various proinflammatory cytokines, including IL-1β, IL-6, IL-8, and TNF-α [22,29]. Serum concentrations of IL-26 are reportedly increased in patients with Crohn’s disease, psoriasis, and rheumatoid arthritis. However, the direct effect of IL-26 on HACs remains unclear. To the best of our knowledge, this is the first report of IL-26 directly inducing expression of COX-2, IL-6, and MMP-1 in HACs.

The pathogenesis of saturated FFAs contributing to obesity-association inflammation has been increasingly elucidated [30]. Palmitate has been shown to induce the expression of COX-2, inflammatory cytokines (IL-1β, TNF-α, IL-6, and IL-8), and MMPs in various cell types [31,32,33,34]. The results of the present study were consistent with those of previous reports that palmitate induced the expression of proinflammatory cytokines and matrix proteinase in HACs [35]. IL-6 expression in HACs was induced by 0.25 mM palmitate in a previous study and by 0.5 mM palmitate in the present study. Palmitate at a concentration of 0.5 mM increased COX-2 mRNA expression in normal HACs [35]. However, in the present study, 0.25 mM palmitate tended to increase COX-2 protein levels, but the changes were not statistically significant. Hence, further studies are needed to investigate the ability of palmitate to induce expression of COX-2. Nonetheless, the results of the present and previous studies showed that the ability of palmitate to induce inflammation of HACs is much weaker than that of IL-26 or the combination of IL-26 and palmitate. The current study supports the idea that obesity, often associated with higher circulating saturated FFA levels, can enhance joint destruction in patients with inflammatory arthritis and high IL26 levels.

COL-II in articular cartilage, which is produced by HACs, provides the capacity to withstand tensile and shear forces. A previous study revealed that 0.5 mM palmitate had no effect on COL-II mRNA levels in HACs [35]. In the present study, 100 ng/mL of IL-26 and 0.25 mM palmitate alone had no effect on COL-II expression in HACs. However, co-treatment with IL-26 and palmitate decreased COL-II formation, which could worsen the metabolic imbalance of articular cartilage.

The results of the present and previous studies showed that IL-26 activated signaling by STAT1, ERK1/2, JNK, and p38 [2,3]. Palmitate has also been shown to activate MAPKs (ERK1/2, p38, and JNK) and the transcriptional factors NF-κB/AP-1 via activation of TLR2/4 in various cell types [31]. However, in the present study, 0.25 mM palmitate activated ERK1/2 in HACs, but not STAT1, STAT3, JNK, or p38. The combination of IL-26 and palmitate for 1 h had a synergistic effect on the activation of ERK1/2 and c-Jun. Inhibitors of TLR4, ERK1/2, and c-Jun were used to confirm the ability of these factors to activate signaling pathways. A previous study revealed that palmitate enhanced TNFα-induced activation of chemokine (C-C motif) ligand 2 via the TLR4-TRIF-IRF-3 signaling cascade in adipose tissue macrophages [36]. However, the results of the present study found no significant change in activation of the TRIF-IRF-3 pathway in HACs (data not shown).

Metformin is an oral hypoglycemic agent that is widely used for the treatment of type 2 diabetes and polycystic ovary syndrome [37]. Several cohort studies have revealed that metformin therapy can decrease body weight [38]. In a prospective cohort study, metformin had a beneficial effect against arthritis in obese patients [24]. A nationwide, retrospective, matched-cohort study showed that metformin combined with COX-2 inhibitors reduced the risk of joint replacement surgery by 25% over a period of 10 years as compared with COX-2 inhibitors only [39]. Metformin was also found to regulate AMPK/mTOR signaling and attenuate the TLR4 inflammatory signaling pathway [25]. In agreement with previous reports, metformin inhibited the synergistic effects of IL-26 and palmitate. Metformin is a very common, low-price medicine. Thus, further studies are warranted to elucidate the mechanism underlying IL-26- and palmitate-induced cartilage damage.

A major limitation to the present study is the origin of HACs. A previous study reported that palmitate at concentrations of <1 mM for 72 h promoted the viability of normal HACs [27]. However, the MTT results showed that 1 mM palmitate for 24 h decreased the viability of HACs. Thus, HACs from osteoarthritic cartilage were used in these experiments. Further studies are needed to investigate these effects in HACS from healthy individuals as well as patients with inflammatory arthritis.

## 5. Conclusions

The present study is the first to report the synergistic effects of IL-26 and palmitate to produce proinflammatory and catabolic mediators in HACs via the TLR4-ERK1/2-c-Jun signaling pathway, which resulted in the loss of the extracellular matrix of articular cartilage (Figure 6). These findings provide a possible mechanism and therapeutic targets to prevent destruction of the articular cartilage in obese patients with inflammatory arthritis.

## Figures and Tables

**Figure 1 cells-10-02500-f001:**
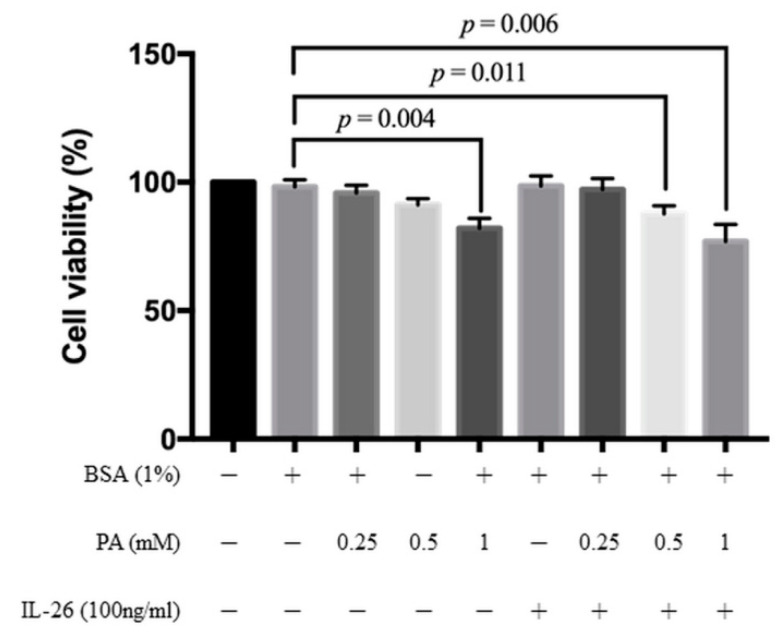
Viability of HACs after exposure to palmitate. The viability of primary HACs treated with palmitate (0.25, 0.5, 1 mM) and with/without IL-26 (100 ng/mL) for 24 h, as determined with the MTT assay. Data are presented as the mean ± SD. Significant differences were detected between the control and palmitate alone or palmitate + IL-26 groups (*n* = 3).

**Figure 2 cells-10-02500-f002:**
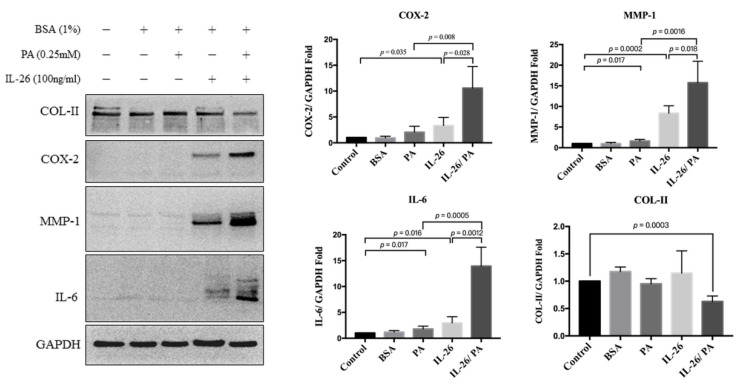
Palmitate and IL-26 synergistically induced the production of inflammatory factors by HACs. Primary HACs were treated with 0.25 mM palmitate alone, 100 ng/mL of IL-26, or a combination of both for 24 h. Total protein was extracted, and inflammatory or extracellular matrix proteins (COX-2, IL-6, MMP-1, and COL-II) were quantified by Western blot analysis. Data are presented as the mean ± SD. Significant differences were detected between the control and palmitate and/or IL-26 groups (*n* = 3).

**Figure 3 cells-10-02500-f003:**
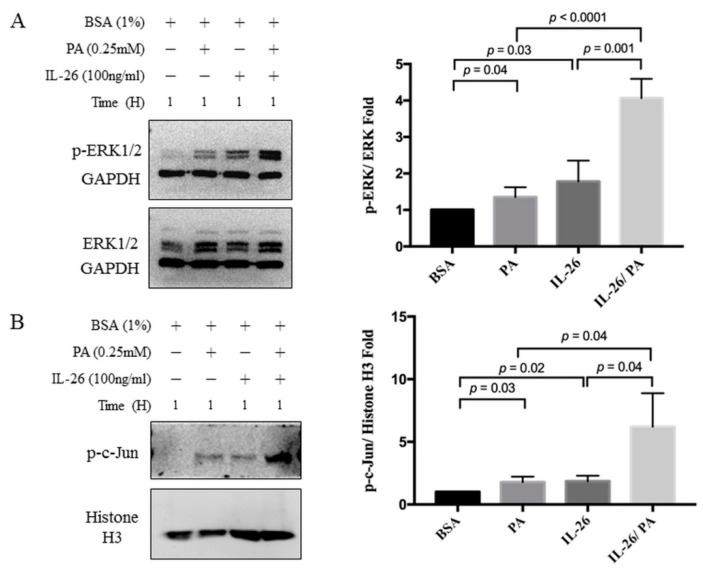
Palmitate and IL-26 synergistically induced activation of inflammatory signaling pathways in HACs. Primary HACs were treated with 0.25 mM palmitate and/or 100 ng/mL of IL-26 for 1 h followed by cytoplasm (**A**) and nuclear extraction (**B**). Data are presented as the mean ± SD. Significant differences were detected between the BSA and palmitate and/or IL-26 groups (*n* = 3).

**Figure 4 cells-10-02500-f004:**
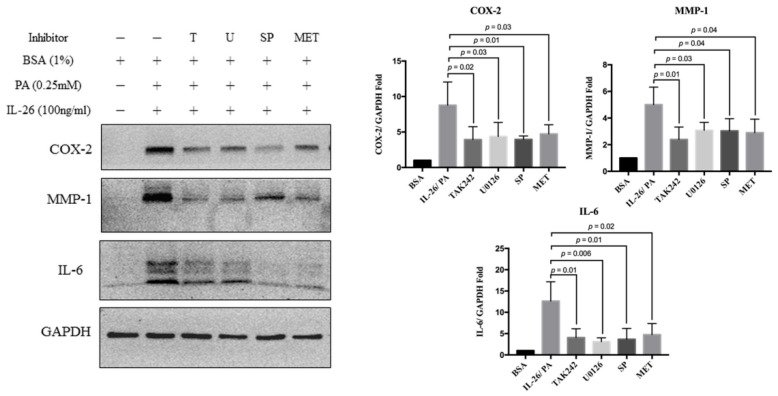
Effects of specific signaling inhibitors and metformin on HACs treated with palmitate and IL-26. COX-2, MMP-1, and IL-6 protein levels in HACs treated with palmitate (0.25 mM), IL-26 (100 ng/mL), TAK242 (10 μM), U0126 (10 μM), SP600125 (25 μM), or metformin (5 mM) for 24 h, as determined by Western blot analysis. Data are presented as the mean ± SD. Significant differences were detected between the IL-26 + palmitate and inhibitor or metformin groups (*n* = 3).

**Figure 5 cells-10-02500-f005:**
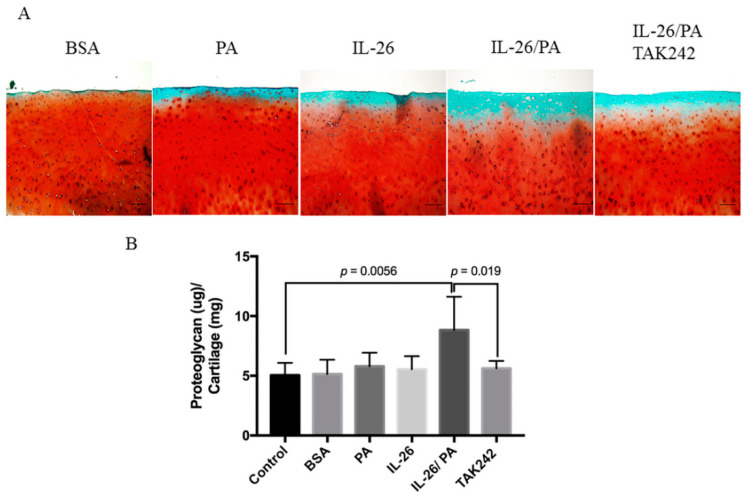
Effects of IL-26 and palmitate on human cartilage explants. Full-thickness cartilage explants were treated with palmitate (0.25 mM) and/or IL-26 (100 ng/mL) with TAK242 (10 μM) for 72 h. (**A**) Safranin O/fast green staining of glycosaminoglycans (GAGs). Magnification: 100×. (**B**) GAG release into the supernatants as determined with the DMMB assay. Images are representative of two different experiments performed in triplicate. Values are expressed as the mean ± SD. (*n* = 6).

**Figure 6 cells-10-02500-f006:**
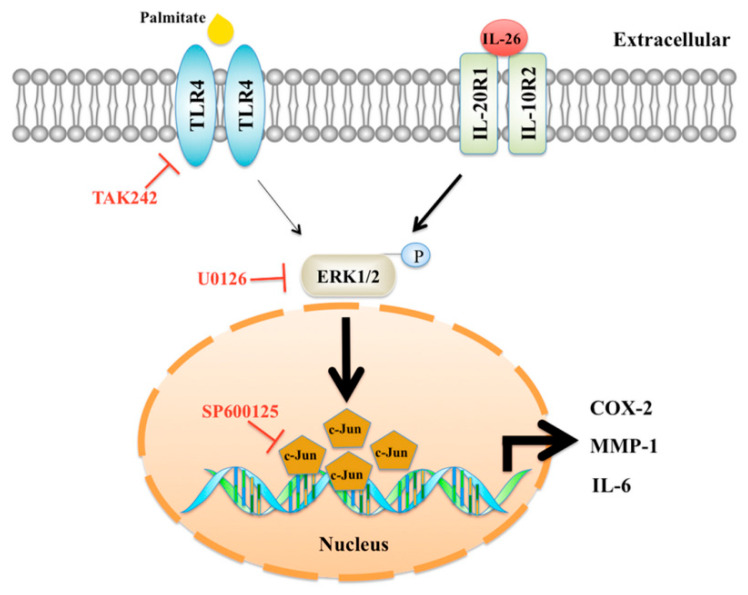
Schematic illustration of signaling pathways underlying the synergy between palmitate and IL-26 for inflammatory mediators. Activation of the synergistic pathways was induced by palmitate and IL-26 via the TLR4-ERK1/2-c-Jun pathway. TAK242 and U0126 inhibited TLR4 internalization and suppressed the synergistic effect of palmitate and IL-26.

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
