# Peer review of "Interleukin-26 Has Synergistic Catabolic Effects with Palmitate in Human Articular Chondrocytes via the TLR4-ERK1/2-c-Jun Signaling Pathway"

_cells, 2021, doi:10.3390/cells10092500_

Round 1

Reviewer 1 Report

Studies on HAC involving the stimulation with proinflammatory cytokines in combination with other factors contributing to metabolic disorders like obesity are very important in the search for the molecular pathways of etiopathogenesis of articular cartilage. I find this study as complete.

Presented study evaluate the negative effects of IL-26 and palmitate in human articular chondrocytes (HAC). The rationale of this research is that interleukin 26 was found to be markedly expressed in synovial fluid upon inflammatory conditions of articular cartilage. Moreover such proinflammatory effects might be enhanced in patients with obese disorders.
First part of methodology included expansion of chondrocytes released enzymatically from articular cartilage until 3 to 5th passage, pre-treatment with inhibitors and stimulation with palmitate with or without addition of IL-26. Second part comprised the range of western blot experiments to show the palmitate and IL-26 effect on the marker expression in the proteins extracted from the HAC culture (The expression levels of COX-2, MMP-1, IL-6, and COL-II). The third part of methodology included evaluation of the catabolic effect of IL-26 and palmitate on articular cartilage relying on the assessment of the degradation level of articular cartilage explants (DMMB and GAG staining of the cartilage explants to show the level of released proteoglycan into the medium).
Upon applied research the authors have shown that combination of IL-26 and palmitate can accelerate extracellular matrix destruction (through the activation of protein markers of ERK1/2-c-Jun signal transduction pathway). The authors revealed that inhibitors of TLR4/ERK1/2/c-Jun signaling pathway efficiently attenuated the catabolic effect of the co-treatment of IL-26 and palmitate in HACs (reduction of increased expression of relevant protein markers). Mentioned results were also confirmed by the authors in articular cartilage using appropriate inhibitor attenuating the palmitate- and IL-26-Induced Matrix Degradation in cartilage explants.
In the conclusion, the authors underlined the combining effect of palmitate and IL-26 on the increase of inflammatory and catabolic mediators of TLR4-ERK1/2-c-Jun signaling pathway in human articular chondrocytes. 

Obtained results of conducted research are novel and important for the pathobiology of articular cartilage. Moreover one of the odds of presented research is the thorough conduction of experiments (like for example the description of results of the assessment of chondrocyte viability and the search for proper concentration of applied adjuvants of the HAC culture). I did not find any weaknesses of them.

Author Response

Thank you for your comments.

Reviewer 2 Report

Very nice job, and important to the field, well done!

Author Response

Thank you for your comments.

Reviewer 3 Report

The study by Yi-Ting Chen et al. aimed to to evaluate the effects of IL-26 and palmitate in human articular chondrocytes. The authors observed the synergistic effects of IL-26 and palmitate in production of proinflammatory and catabolic mediators in HACs via the TLR4-ERK1/2-c-Jun signaling pathway. In consequence it resulted in the loss of the extracellular matrix of articular cartilage. The finding may be of special importance for obese patients with inflammatory arthritis to prevent destruction of the articular cartilage. The study is well designed and written. However, some issues should be raised:

  1. The p values should be shown in the figures.
  2. The quality of the figures should be improved, especially Fig. 2-6.
  3. To compare mean values between the groups (control group, palmitate alone or palmitate+IL-26), I would recommend to use nonparametric test as n is 3 or 6 in your experiment.

Author Response

Many thanks for the reviewer's thoughtful comments and helpful suggestions. All the comments have been carefully considered and responded on a point-by-point basis. The critically corrected parts in the revised manuscript were marked it by "Track Changes".

point 1. The p values should be shown in the figures.

Response: Thank you for your suggestion, we have shown the p values in all figures.

Point 2. The quality of the figures should be improved, especially Fig. 2-6.

Response: Thank you for your suggestion, we have updated figures at revised version.

Point 3. To compare mean values between the groups (control group, palmitate alone or palmitate+IL-26), I would recommend to use nonparametric test as n is 3 or 6 in your experiment.

Response: Thank you for your suggestion. Following your suggestion to analyze our data by nonparametric test, the statistical differences between groups cannot be revealed. Thus, we still analyze our data by unpaired parametric t-test to tell the difference between groups.
